# CAR Based Immunotherapy of Solid Tumours—A Clinically Based Review of Target Antigens

**DOI:** 10.3390/biology12020287

**Published:** 2023-02-10

**Authors:** John Maher, David M. Davies

**Affiliations:** 1CAR Mechanics Group, Guy’s Cancer Centre, School of Cancer and Pharmaceutical Sciences, King’s College London, Great Maze Pond, London SE1 9RT, UK; 2Department of Immunology, Eastbourne Hospital, Kings Drive, Eastbourne BN21 2UD, UK; 3Leucid Bio Ltd., Guy’s Hospital, Great Maze Pond, London SE1 9RT, UK

**Keywords:** solid tumour, chimeric antigen receptor, T-cell, NK cell, target, toxicity

## Abstract

**Simple Summary:**

Cancer accounts for an increasing number of deaths year on year. However, new immune-based therapies offer promise in the quest to address this unmet need. One approach entails the use of immune white blood cells (usually collected from the patient themselves) and introducing a genetic blueprint that enables these cells to identify and attack cancer cells. These so-called “CAR cells” have proven to be very effective in the treatment of blood cancers. However, solid tumours (which account for 90% of all cancers) are proving more difficult to treat with this approach. One of the key challenges is identifying targets that clearly distinguish between cancer cells and normal, healthy tissue. Here, we have surveyed targets that have been selected for clinical testing of CAR cells in the context of clinical trials.

**Abstract:**

Immunotherapy with CAR-engineered immune cells has transformed the management of selected haematological cancers. However, solid tumours have proven much more difficult to control using this emerging therapeutic modality. In this review, we survey the clinical impact of solid tumour CAR-based immunotherapy, focusing on specific targets across a range of disease indications Among the many candidates which have been the subject of non-clinical CAR T-cell research, clinical data are available for studies involving 30 of these targets. Here, we map out this clinical experience, highlighting challenges such as immunogenicity and on-target off-tumour toxicity, an issue that has been both unexpected and devastating in some cases. We also summarise how regional delivery and repeated dosing have been used in an effort to enhance impact and safety. Finally, we consider how emerging armouring systems and multi-targeted CAR approaches might be used to enhance tumour access and better enable discrimination between healthy and transformed cell types.

## 1. Introduction

Chimeric antigen receptors are synthetic polypeptides in which a target-specific binding moiety is fused to a spacer and membrane-spanning element, followed in turn by a bespoke signalling domain. Second-generation CAR T-cells that incorporate both a co-stimulatory and activating module have proven transformative in the management of selected haematological malignancies [1,2] (Figure 1). By contrast, success against solid tumours has been much more modest, owing to the multitude of additional hurdles imposed in that setting [3,4]. The first challenge concerns the lack of tumour-specific targets, meaning that the risk of on-target off-tumour toxicity is a pressing concern. This issue has been repeatedly underlined when novel CAR products have been evaluated in man for the first time [5,6]. Given the potential use of CAR-engineered immune cells for the treatment of multiple distinct tumour types in which target expression is shared, the focus of this review is to summarise and appraise the clinical data arising from CAR-based solid tumour immunotherapy studies and case reports. Target selection in currently registered clinical trials worldwide is indicated in Figure 2. In a partner review article, we surveyed published pre-clinical experience of CAR-based immunotherapy directed at solid tumour targets where clinical data are not available at this time (Maher, J. and Davies, D.M.. Cancers (2023) In press).

## 2. Mesothelin

Figure 1 indicates that mesothelin has been the most popular target antigen evaluated in CAR-T-cell clinical trials involving solid tumours. Mesothelin is expressed on a range of solid tumours including pancreatic, ovarian and lung carcinoma and malignant pleural mesothelioma. However, mesothelin is also expressed in normal mesothelial cells found in the pleura, pericardium and peritoneum. To mitigate the risk of on-target off-tumour toxicity, initial clinical testing conducted at the University of Pennsylvania involved the manufacture of engineered T-cells in which the CAR was transiently expressed by mRNA transfection [7,8]. Good safety was demonstrated with the exception of a single incidence of anaphylaxis following CAR T-cell re-treatment [9]. Next, a follow-on study was undertaken using lentiviral transduced (e.g., stably modified) CAR T-cells [10]. Five patients each with pancreatic cancer, ovarian cancer or malignant pleural mesothelioma were treated, and the best overall response was stable disease. One patient died of sepsis, and anti-CAR antibodies were detected in a number of subjects, presumably due to the inclusion of a murine scFv in the CAR ectodomain. To address this, a human scFv was next incorporated to reduce the immunogenicity of the CAR. Fourteen patients with mesothelin-expressing tumours were treated in this clinical study [11]. Once again, the best response was stable disease. One patient died as a result of respiratory failure, and investigators demonstrated that mesothelin was upregulated in areas of inflammation and fibrosis within the lung, suggesting that on-target off-tumour toxicity was responsible for this event (https://www.med.upenn.edu/cellicon2021/assets/user-content/documents/tanyi.pdf, accessed on 15 December 2022).

An early clinical trial of mesothelin-specific CAR T-cells was also undertaken at the National Cancer Institute. While safety was acceptable, only 1 of 15 patients achieved stable disease [12]. Investigators at Memorial Sloan Kettering Cancer Center set out to boost both the efficacy and safety of this approach through regional (e.g., intrapleural) delivery of these cells in patients with malignant disease at that site (mainly mesothelioma) [13]. Treatment using this localized delivery approach was safe, even with repeated dosing, with no dose-limiting toxicities. Moreover, two patients who also received the anti-PD1 antibody pembrolizumab and cyclophosphamide conditioning achieved partial responses (PR) by RECIST criteria, both of which were accompanied by complete metabolic responses, as detected using positron emission tomography (PET) using ^18^fluoro-deoxyglucose (FDG). To further potentiate this approach, investigators co-expressed a dominant-negative PD1 receptor and employed a calibrated CD3ζ endodomain within the CAR in which ITAMs 2 and 3 are inactivated [14,15]. Unfortunately, however, despite the use of regional delivery, a fatal serious adverse event has occurred in this trial, prompting a pause to enrolment (https://investors.atarabio.com/news-events/press-releases/detail/265/atara-biotherapeutics-provides-update-on-ata2271-autologous, accessed on 15 December 2022). At the time of writing (27 January 2023), this trial (NCT04577326) is listed as recruiting.

Regional (e.g., intraperitoneal) delivery has also been employed by MaxCyte who have engineered peripheral blood mononuclear cells by mRNA transfection to express a mesothelin-specific CAR, allowing for rapid manufacture and repeated dosing. When last reported in a clinical trial involving patients with ovarian and peritoneal mesothelioma, stable disease was the best result achieved, while tolerability was good [16]. Mesothelin-specific CAR T-cells have also been evaluated in three patients with epithelial ovarian cancer [17]. Although therapy-related toxicity was acceptable, one patient died unexpectedly 200 days after receiving the CAR T-cells, having experienced some tumour reduction. A second patient achieved disease stabilization while the third treated subject progressed following treatment.

To boost tumour infiltration and survival of the CAR T-cells, another approach that has been evaluated entails the repeated administration of mesothelin-specific CAR T-cells, armoured using both IL-7 and CCL19. Cells were delivered via the hepatic artery (dose 1) or intravenously (all other doses) [18]. One patient with pancreatic cancer received 5 doses, and complete tumour remission was achieved. A second patient treated in this study had advanced ovarian carcinoma and progressed following intraperitoneal delivery of armoured mesothelin-specific CAR T-cells. Further updates from this study are awaited.

Finally, TCR^2^ Therapeutics are evaluating a novel CAR design in which a mesothelin-specific scFv is directly fused to CD3ε, thereby incorporating the CAR into the endogenous T-cell receptor/CD3 complex (https://www.onclive.com/view/gavo-cel-elicits-clinical-benefit-in-solid-tumors-including-ovarian-cancer-and-mesothelioma, accessed on 15 December 2022). In phase 1 testing, they have recently announced an overall response rate of 22% in patients with mesothelin-expressing malignancies. Among 32 treated patients, 2 dose-limiting toxicities (DLTs) were encountered, namely grade 3 pneumonitis (which responded to anti-cytokine therapy) and a fatal bronchoalveolar haemorrhage.

Clinical trials of mesothelin targeted CAR technologies known to be currently recruiting are listed in Table 1.

## 3. Receptor Tyrosine Kinases

### 3.1. HER2

HER2 is a receptor tyrosine kinase and a bona fide driver of human cancer. Over-expression of HER2 due to gene amplification and/or transcriptional dysregulation is observed in about 20% of breast cancers, a smaller proportion of gastric/gastro-oesophageal junction cancers and, less commonly, in other solid tumour types. Given the clinical success of HER2-targeted monoclonal antibodies such as traztuzumab in the treatment of HER2 over-expressing cancers [19], it was logical that it would become a target of interest for CAR T-cell development. Indeed, HER2 was first the subject of CAR T-cell targeting in 1993 [20]. Since then, there has been extensive pre-clinical evaluation of CAR T-cells directed against HER2, as reviewed in [21].

Clinical testing of HER2 targeted T-cells has yielded mixed results. The safety of this approach was profoundly questioned by an early case report in which a patient with HER2+ colorectal cancer was treated with 10 billion third-generation CAR T-cells, infused intravenously after lymphodepleting chemotherapy. The patient died after 5 days due to on-target off-tumour toxicity, variously attributed to recognition of low levels of HER2 in the pulmonary parenchyma [6] or vasculature [22]. Subsequently, however, the principle that HER2 targeted CAR T-cells could be safely administered systemically was demonstrated in sarcoma patients at doses of up to 1 × 10^8^ cells/m^2^, infused initially without lymphodepletion [23]. Importantly, these tumours expressed HER2 at levels below those seen in HER2-amplified tumours [24]. Evidence of CAR T-cell trafficking to tumours was presented, accompanied by 1 partial response (PR) achieved after a second dose of CAR T-cells. More recently, 10 HER2+ sarcoma patients received up to three infusions of 1 × 10^8^ CAR T-cells/m^2^, each administered after lymphodepletion followed by up to 5 subsequent infusions without lymphodepletion (NCT00902044) [25]. Infusions were well tolerated, with expected cytopenias and cytokine release syndrome (CRS) that did not exceed grade 2. CAR T-cells expanded with a median peak at day 7 and they remained detectable in the circulation for 6 weeks. One patient with rhabdomyosarcoma (HER2+ at grade 3 and intensity score 3) received three 10 weekly cycles of lymphodepletion with fludarabine and cyclophosphamide followed by 1 × 10^8^ CAR T-cells/m^2^ [26]. This induction therapy resulted in a complete response (CR), which was confirmed morphologically in bilateral bone marrow aspirates from cycle 3 onwards, with stable disease detected after cycles 1 and 2. FDG-PET scanning confirmed a complete metabolic response upon completion of cycle 3 of induction therapy. Treatment was well tolerated with expected cytopenias and grade 1 CRS but no evidence of on-target off-tumour toxicity. Four additional cycles of consolidation CAR T-cell immunotherapy were administered, and CR was maintained for a total of 12 months. Evidence of T-cell receptor re-modelling in the peripheral blood and autoantibody responses against tumour-associated proteins supported the possibility that an endogenous anti-tumour immune response had been activated by CAR T-cell treatment. At the time of disease relapse, HER2 expression was maintained on the malignant cells, albeit at lower levels than previously. Nonetheless, CR was re-induced following the administration of a single cycle of CAR T-cells post lymphodepletion. CAR T-cells were demonstrated in bone marrow at the time that remission was demonstrated. Two further induction cycles (with lymphodepletion) followed by five consolidation cycles of CAR T-cells alone were infused, and the child remained in CR at 20 months after completion of CAR T-cell immunotherapy and 4.3 years after receiving the first of these infusions. These data establish important proof of concept for the potential utility of repeated CAR T-cell infusions in patients with refractory solid tumours. 

HER2-targeted CAR T-cells have also been evaluated in patients with biliary tract and pancreatic tumours (NCT01935843) [27]. Patients received 1 or 2 doses of CAR T-cells following conditioning with nab-paclitaxel and cyclophosphamide. Among eleven treated patients, one partial response of 4.5 duration was recorded. However, two cases of upper gastrointestinal haemorrhage were reported within the dose-limiting toxicity period, raising concerns once again about on-target off-tumour toxicity.

Brain tumours may provide a particular clinical niche for the deployment of HER2-targeted CAR T-cells. While HER2 is undetectable in normal brain tissue, it is expressed in a range of brain tumours, including glioblastoma [28], ependymoma [29] and medulloblastoma [29,30]. To target this, Ahmed et al. used virus-specific cytotoxic T-cells engineered to express a HER2-specific CAR [28]. One or more doses were administered intravenously without lymphodepletion. CAR T-cells did not expand but nonetheless remained detectable in the peripheral blood for up to 12 months, and no dose-limiting toxicities were observed. One PR was observed for over 9 months, while stable disease (SD) was seen in seven additional patients for up to 29 months. Repeated dosing of HER2-targeted CAR T-cells via a central nervous system (CNS) catheter has also been undertaken in children and young adults with relapsed refractory brain tumours [31], highlighting the potential of this approach to trigger a local inflammatory response. 

An important consideration regarding the targeting of HER2 using CAR T-cells pertains to the level of HER2 expression required on tumour cells for effective targeting. In the US, HER2+ breast cancer is defined by the demonstration of complete and intense circumferential HER2 protein membrane staining in >10% of tumour cells at a 3+ score using immunohistochemistry (IHC) and/or by the demonstration of amplification of the HER2 gene [32]. In this circumstance, HER2-targeted monoclonal antibody therapies such as traztuzumab and pertuzumab are recommended, together with taxane chemotherapy. Recently, however, the antibody–drug conjugate, trastuzumab deruxtecan, has proven highly efficacious in patients with relapsed refractory breast cancer in which HER2 expression is present, but at levels below those required to define the tumour as HER2+ [33]. This seminal clinical study establishes the precedent that HER2-targeted therapies may benefit patients with solid tumours that do not meet the definition of HER2+ disease. In the context of breast cancer, approximately 50% of tumours fall into this category, including some triple-negative breast tumours. Several additional solid tumour types express HER2, but at levels not considered to recommend treatment with traditional HER2-targeted therapies. This development may also have a significant impact on the development of CAR T-cell therapies that recognise lower levels of HER2. While trastuzumab deruxtecan will always be favoured over a CAR T-cell approach owing to ease of manufacture/delivery and lower cost, the curative potential of this drug is likely to be low. Illustrating this, in previously treated HER2-low advanced breast cancer, trastuzumab deruxtecan achieved a complete response rate of 3.6%, accompanied by a median progression-free and overall survival of 9.9 and 23.4 months, respectively [33]. These considerations provide a rationale for the future positioning of efficacious HER2-targeted CAR T-cell products as salvage therapy when existing biological agents such as trastuzumab deruxtecan fail. They also raise the prospect that HER2-targeted therapies, including CAR T-cell immunotherapy, may find application in the treatment of a much greater spectrum of solid tumours than is currently the case. Clinical trials of HER2-targeted CAR technologies known to be currently recruiting are listed in Table 2.

### 3.2. Epidermal Growth Factor Receptor (EGFR)

Epidermal growth factor receptor (EGFR) is widely over-expressed in epithelial-derived solid tumours. The greatest clinical experience with EGFR-targeted CAR T-cell immunotherapy has been achieved by investigators at the PLA General Hospital, Beijing, China. In an initial study, 11 patients with relapsed refractory non-small cell lung cancer (NSCLC) in which >50% tumour cells were EGFR+ were infused with up to 2.5 × 10^7^ CAR T-cells/kg [34]. Two partial responses were seen, although both patients had received conditioning therapy (including cisplatin, cyclophosphamide and either pemetrexed or docetaxel), which may have influenced outcome. Stable disease was reported in a further five patients. Therapy was well tolerated with the exception of a grade 3–4 increase in serum lipase. A more recent study in NSCLC was undertaken by a group in Shanghai using PiggyBac transposon-engineered T-cells, administered as two doses. Infusions were generally well tolerated, and one PR was reported in nine patients.

In a follow-up study, the Beijing group tested EGFR-targeted CAR T-cells in patients with biliary cancers that contained >50% EGFR+ cells [35]. They employed nab-paclitaxel and cyclophosphamide conditioning in order to achieve lymphodepletion, while simultaneously disrupting the tumour-associated stroma. Cells were infused in fractionated doses (for safety reasons) over 3–5 days at doses of up to 4 × 10^6^ CAR T-cells per kg (mean transduction efficiency, 9%). CAR T-cell re-dosing occurred in some patients on one or two occasions. Of the 17 evaluable patients, there was 1 CR and 10 cases of disease stabilization. Further CR and PR were achieved in two patients who did not receive conditioning therapy. Several patients experienced grade 1–2 epithelial toxicities such as mucositis, desquamation and gastrointestinal haemorrhage, while one patient developed acute pulmonary oedema that responded to the anti-IL-6 receptor antibody, tocilizumab.

The same group also reported a clinical trial in patients with pancreatic cancer [36]. The required level of tumour cell EGFR positivity, CAR T-cell dosing/re-dosing and conditioning chemotherapy were all similar to those in the biliary cancer study. Toxicity included several cases of mucosal and cutaneous toxicities in addition to two cases of pulmonary oedema/pleural effusions and one gastrointestinal haemorrhage. Four PRs were observed in fourteen evaluable patients. All studies reported a correlation between response and the proportion of central memory cells present in the infused product, although it is unclear if this pertains to the transduced cells only. Infiltration of CAR T-cells into tumour deposits was also demonstrated.

One novel approach to targeting EGFR entails the use of a CAR that binds to an epitope that is not expressed in normal EGFR-expressing cells (irrespective of ligand binding), but is exposed when EGFR is over-expressed, truncated or mutated. A phase 1 clinical trial involving such a CAR is currently underway in children and young adults with EGFR-expressing solid tumours [37]. A dose-limiting toxicity of abnormal liver function tests was noted at the second planned dose level of 1 × 10^6^ cells/kg. Three mixed responses have been noted to date.

An alternative strategy to target EGFR entails the use of the chimeric T1E polypeptide, which is a promiscuous ErbB ligand that directs CAR T-cell specificity against all EGFR homo- and heterodimers in addition to ErbB4 homodimers and both ErbB2/3 and ErbB2/4 heterodimers [38]. CAR T-cells directed using this ligand are undergoing evaluation by intratumoural delivery in patients with head and neck cancer. Treatment was well tolerated, and stable disease was achieved in 9 of 15 patients treated during the dose-escalation phase of the study [39].

Clinical trials of EGFR targeted CAR T-cells known to be currently recruiting are listed in Table 3.

### 3.3. Epidermal Growth Factor Receptor Variant III (EGFRvIII)

EGFRvIII is a tumour-specific splice variant of EGFR in which amino acids 6–273 of the extracellular domain are deleted, resulting in constitutive receptor activity. It is expressed in about 30% of glioblastoma multiforme (GBM) tumours, in addition to a smaller number of other tumour types. Although not found in normal tissues, expression is typically heterogeneous within the malignant cell population.

Evaluation of intravenously administered EGFRvIII-specific CAR T-cells was undertaken in a cohort of 10 patients with EGFRvIII+ GBM [40]. Efficacy could not easily be assessed using imaging owing to difficulties in distinguishing inflammatory reactions from disease progression. Nonetheless, one patient remained clinically stable for 18 months. Evidence of the trafficking of CAR T-cells to the tumour was noted, accompanied by an influx of other T-cells, many of which had regulatory properties. Other changes observed in the tumour microenvironment post-CAR T-cell infusion included the upregulation of PD-L1, indoleamine dioxygenase and IL-10. In most cases, tumours that were resected post-infusion demonstrated lowered expression of EGFRvIII, consistent with antigen loss as a CAR T-cell evasion mechanism. Expression of PD1 on the infused cell product was linked to enhanced persistence and therapeutic response [41].

Clinical trials of EGFRvIII targeted CAR T-cells known to be currently recruiting are listed in Table 4.

### 3.4. Receptor Tyrosine Kinase-like Orphan Receptor Family Member (ROR)1

ROR family members are Wnt ligand-binding receptor tyrosine kinases and are mainly expressed during embryonic development. Expression of ROR1 has also been reported in a range of malignancies, including chronic lymphocytic leukaemia, mantle cell leukaemia, some myeloid leukaemias, melanoma, triple-negative breast cancer, neuroblastoma, non-small cell lung cancer and tumours derived from the ovary, bowel, stomach, pancreas, prostate, kidney and endometrium [42]. Low-level expression of ROR1 has also been described in adult regenerating B-cells (haematogones), parathyroid gland, pancreatic islets, adipose tissue and the gastrointestinal tract [43]. In addition, expression is increased by chronic inflammation and fibrosis [44]. Safety of ROR1-specific CAR T-cells has been demonstrated in non-human primates [45]. Moreover, a recent study of a ROR1-specific antibody–drug conjugate also provided support for the safety of this target [46]. On the other hand, ROR1-specific CAR T-cells have caused severe bone marrow failure in mice due to stromal cell targeting when administered after radiation to induce lymphodepletion [47]. This could be overcome using synthetic (Syn)Notch technology, whereby expression of the ROR1 CAR was driven by recognition of a target (B7-H3 or EpCAM) that was found on tumour cells but not stromal cells [47]. Alternatively, universal CAR T-cells may be co-administered with a bispecific adaptor protein that directs specificity against ROR1, allowing clinician-controlled delivery of therapeutic impact [48].

In parallel with these pre-clinical developments, a clinical trial of ROR1-targeted CAR T-cells in patients with NSCLC and triple-negative breast cancer (TNBC) was initiated at the Fred Hutchinson Cancer Center (NCT02706392) [49]. No significant toxicity was noted in the first three treated patients. However, poor trafficking to tumours was noted, accompanied by a lack of efficacy. Six patients were ultimately treated in this study without any DLTs, although the trial was terminated owing to slow patient accrual [50]. This prompted investigators to explore the use of oxaliplatin chemotherapy in pre-clinical models to enhance CAR T-cell tumour trafficking [49]. A second trial of ROR1-specific CAR T-cells is ongoing in adults with TNBC and NSCLC, sponsored by Lyell Immunopharma (NCT05274451).

### 3.5. c-MET (Hepatocyte Growth Factor Receptor)

The c-MET receptor tyrosine kinase is the receptor for hepatocyte growth factor and is expressed in a range of solid tumours, including liver, lung, breast, prostate, pancreas and gastrointestinal tract carcinomas, glioblastoma and malignant mesothelioma. It is pro-tumourigenic, favouring epithelial-to-mesenchymal transition and ultimately conferring a worsened prognosis. However, c-MET is expressed in a number of normal tissues also (https://www.proteinatlas.org/ENSG00000105976-MET/tissue, accessed on 22 December 2022). Consequently, clinical evaluation of c-MET-targeted CAR T-cells was initially undertaken using mRNA-transfected T-cells, which were administered using the intratumoural route (cutaneous or lymph node metastases) to six patients with metastatic breast cancer [51]. Treatment was well tolerated in all cases. In some cases, low-level leakage of CAR T-cells into the peripheral blood was detectable soon after administration. Injected lesions were surgically excised after 48 h, revealing necrosis and c-MET downregulation. No ongoing clinical trials of CAR-based immunotherapy directed at c-MET were identified on clinicaltrials.gov (accessed on 22 December 2022).

### 3.6. Vascular Endothelial Growth Factor Receptor (VEGFR) 2

Vascular endothelial growth factor receptor (VEGFR) 2 is expressed in tumour-associated vasculature. CAR T-cell immunotherapy using VEGFR 2-targeted cells was evaluated in a clinical trial undertaken at the National Cancer Institute. Twenty-four patients with melanoma were treated and also received IL-2. The study was terminated because no responses were seen and five serious adverse events were reported (https://clinicaltrials.gov/ct2/show/results/NCT01218867, accessed on 22 December 2022). No ongoing CAR trials directed against this target were identified on clinicaltrials.gov (accessed on 22 December 2022).

## 4. Mucins

Mucins are heavily glycosylated cell surface molecules, two of which have been targeted clinically using CAR T-cells.

### 4.1. MUC-1 (Mucin-1)

An exercise undertaken by the National Cancer Institute in 2009 ranked MUC1 as the top cell surface tumour antigen for immunotherapeutic targeting (falling in second place for all candidates behind Wilms’ tumour antigen 1) [52]. MUC1 is an attractive target for CAR T-cell immunotherapy for three reasons. First, it is transcriptionally upregulated in many cancers, including those arising from the breast, pancreas and ovary. Second, MUC1 is normally expressed in a polarised and inaccessible manner on the secretory epithelium, but polarity is lost upon malignant transformation, theoretically enhancing its accessibility to CAR T-cells. Perhaps its most attractive attribute is the fact that MUC1 is underglycosylated in many cancers. As a result, antibodies with specificity for the tandemly repeated immunodominant epitope within the MUC1 ectodomain (e.g., HMFG2 or SM3 antibodies) bind preferentially to a number of tumour-associated glycoforms of this mucin, while the 5E5 antibody selectively binds to MUC1 that carries the Tn antigen at this position [53]. Owing to its aberrant glycosylation, tumour-specific CAR T-cells targeted against MUC1 have been described by a number of groups [54,55].

Despite this background, there has only been limited clinical evaluation of MUC1-specific CAR T-cells. A case report highlighted the ability of CAR T-cells targeted with a modified SM3 scFv to cause necrosis of a metastatic MUC1+ lesion in a patient with seminal vesicle carcinoma [56]. A 5E5-targeted CAR is currently undergoing clinical evaluation by the company, Tmunity Therapeutics, in patients with MUC1 Tn-expressing solid tumours [57]. The CAR contains a CD2 co-stimulatory domain with the goal of delaying the onset of T-cell exhaustion. Thus far, treatment has been well tolerated, and stable disease has been achieved in 3 of 6 patients, all of whom had received prior lymphodepletion with fludarabine and cyclophosphamide.

As indicated above, the MUC1 ectodomain is markedly post-transcriptionally altered in transformed compared to healthy cells, offering a unique discriminating factor for targeting purposes. Nonetheless, one potential disadvantage of targeting the tandemly repeated immunodominant epitope within the MUC1 ectodomain is the fact that the latter is both shed and large, extending up to 500 nm from the cell surface. Consequently, some companies such as Minerva Therapeutics and Poseida Therapeutics are taking a distinctive approach by targeting cleaved MUC1 (MUC1*) [58] or the C-terminal subunit of MUC1 respectively [59]. The Poseida approach employs allogeneic T-cells that have been engineered using PiggyBac^®^ transposon technology, which leads to an increase in the proportion of memory stem cells present. Cas-CLOVER^TM^ technology is used to edit out the *B2M* and *TRAC* genes, thereby mitigating the risk of immune rejection and inducing graft versus host disease, respectively. Peer-reviewed data are pending from both trials at the time of writing, although Poseida recently announced that they had treated six patients, one of whom achieved a partial response without significant toxicity (https://poseida.com/wp-content/uploads/2022/12/46P-Oh.pdf, accessed on 12 December 2022). Clinical trials of MUC1-targeted CAR T-cells known to be currently recruiting are listed in Table 5.

### 4.2. MUC-16 (Mucin-16)

MUC16 (also known as CA125) is expressed on respiratory and ocular epithelium in addition to mesothelial surfaces and reproductive organs in both sexes [60]. Expression of this mucin is upregulated in over 80% of ovarian cancers [61], in addition to some pancreatic and endometrial cancers. A phase 1 clinical trial was undertaken in patients with relapsed refractory ovarian cancer in which IL-12 armoured MUC16-specific CAR T-cells were first administered intravenously. If this dose was well tolerated, a second equivalent dose was infused using the intraperitoneal route after 24–48 h. When cells were administered without lymphodepleting chemotherapy, CRS was observed but without DLTs. However, when patients were conditioned with lymphodepleting chemotherapy prior to CAR T-cell infusion, two episodes of macrophage-activation syndrome were observed, both of which constituted DLTs. It is unclear whether this toxicity may be attributable, at least in part, to IL-12 armouring, given the potently pro-inflammatory nature of this cytokine.

### 4.3. Tumour-Associated Glycoprotein 72 (TAG-72)

Tumour-associated glycoprotein 72 (TAG72) is an oncofoetal mucin that is expressed by most adenocarcinomas, albeit heterogeneously and subject to antigenic shedding [62]. Limited expression by normal secretory endometrial tissue and duodenal goblet cells has been reported [63]. Two pioneering clinical trials that evaluated a first-generation CAR targeted against TAG-72 were undertaken in patients with metastatic colorectal cancer, both initiated 25 years ago [62]. These involved repeated dosing using the intravenous (n = 14) or intra-hepatic arterial delivery route (n = 9), thereby achieving preferential delivery to metastatic liver disease. Interferon-α was co-administered in both studies in an effort to further upregulate target antigen expression. However, no clinical responses were observed. Safety data were incomplete, but there were a few cases of low-grade CRS and some unexplained events such as a single episode of retinal artery occlusion. No evidence of on-target off-tumour toxicity was reported. Anti-idiotype antibodies occurred in most patients despite the use of a humanised scFv, giving rise to an artifactual reduction in circulating TAG-72 levels. Imaging of indium-111 labelled CAR T-cells using a gamma counter demonstrated clustering of tracer at the periphery of large tumour masses, consistent with trafficking to, but poor penetration of the CAR T-cells into the tumour core. CAR T-cells generally persisted for 6–14 weeks, although in one case, persistence to 48 weeks was demonstrated. One ongoing TAG-72-specific CAR T-cell clinical trial was identified at City of Hope Medical Center involving patients with platinum-resistant ovarian cancer (NCT05225363).

## 5. Claudins

Members of the claudin protein family contribute to tight junction formation between neighbouring epithelial and endothelial cells, a process that may be disrupted in cancer. Expression of members of the family may be downregulated or upregulated in various tumour types, and two oncofoetal claudins have been targeted in men using CAR T-cell immunotherapy. In both cases, pre-clinical studies have provided support for a “CARVac” strategy whereby a liposomal mRNA vaccine is used to engineer expression of the relevant claudin in antigen-presenting cells, thereby boosting proliferation and persistence of the CAR T-cells [64].

### 5.1. CLDN6 (Claudin-6)

Claudin-6 is highly expressed in a range of foetal tissues but not in their adult counterparts [65]. It is aberrantly expressed in up to 100% of germ cell tumours in addition to ovarian, cervical, gastric and liver carcinomas [65]. An ongoing multicentre phase 1 clinical trial is evaluating CLDN6-specific CAR T-cells in lymphodepleted recipients, administered alone or in combination with repeated CARVac dosing [66,67]. In a preliminary report, partial responses were achieved in 6 of 18 subjects, comprising 4 testicular cancer and 2 ovarian cancer patients. One case of grade 4 macrophage activation syndrome was also reported. A subsequent update indicates that among 21 evaluable patients, there were 6 PRs and 1 CR (in a patient with testicular cancer; https://www.onclive.com/view/cldn6-car-t-cell-therapy-shows-encouraging-efficacy-in-relapsed-refractory-advanced-solid-tumors, accessed on 16 December 2022).

### 5.2. CLDN18.2 (Claudin-18.2)

Claudin-18.2 is the stomach-specific isoform of CLDN18 and is highly expressed in gastro-oesophageal and pancreatic tumours. By contrast, expression in adult healthy tissue is restricted to differentiated cells within the gastric mucosa, which are relatively inaccessible to CAR T-cells. An interim analysis of a phase 1 clinical trial of CLDN18.2-specific T-cells described outcomes in 37 subjects with gastrointestinal (GI) malignancies, including gastric, gastro-oesophageal junction and pancreatic cancer [68]. The most common conditioning regimen used was the combination of fludarabine, cyclophosphamide and nab-paclitaxel. Although no DLTs were noted, there were four grade 3–4 GI toxicities, including one grade 4 GI haemorrhage, prompting a reduction in CAR T-cell dosing thereafter in the trial. The overall response rate was very impressive at 48.6% (57.1% in gastric cancer) with a 6-month overall survival of 80.1%. A second single centre trial was reported in abstract format in 2019 (although the current status of this trial is unknown) [69]. In that study, 12 subjects with gastric or pancreatic cancer were treated with 1–5 cycles of CAR T-cells after lymphodepletion with fludarabine, cyclophosphamide and nab-paclitaxel. Among the 11 evaluable patients, there was 1 CR and 3 PRs, with no DLTs or grade 4 toxicity other than cytopaenias.

Clinical trials of CLDN18 targeted CAR T-cells known to be currently recruiting are listed in Table 6.

## 6. FR (Folate Receptor)-α

Folate receptor (FR)α is expressed on over 80% of ovarian carcinomas in addition to cancers arising in the fallopian tubes, peritoneal cavity, lung, uterus, breast and pleural space. Low-level expression of this receptor is found in polarized epithelia such as kidney, lung, choroid plexus and in haematopoietic tissues. The first clinical trial of CAR T-cell immunotherapy involved the use of first-generation CAR T-cells to treat patients with ovarian cancer [70]. While the safety profile was good, there was no evidence of CAR T-cell trafficking to tumour, and no responses were observed. Clinical trials of FRα targeted CAR T-cells known to be currently recruiting are listed in Table 7.

## 7. IL13Rα2

Interleukin-13 receptor a2 (IL13Rα2) is a high-affinity monomeric receptor for IL-13 and is over-expressed in a broad range of solid tumours. This includes the majority of glioblastomas [71], ovarian cancer (83%) [72], pancreatic ductal adenocarcinoma (71%) [73], colorectal cancer (66%) [74], head and neck cancer (33%) [75], and other tumours such as breast cancer, melanoma, thyroid cancer, mesothelioma and clear cell renal cell carcinoma. Upregulation of IL13Rα2 has been linked to chronic inflammation, which may underlie its frequent expression in these various cancer types [76]. By contrast, expression in normal tissue is largely limited to spermatocytes within the testis [77] and pituitary gland [76], with low transcript levels seen in other normal tissues [78]. Accordingly, it has been argued that IL13Rα2 is a form of cancer-testis antigen [77]. Originally believed to act as a decoy receptor, IL13Rα2 signals via STAT6-independent pathways to promote tumour cell survival, invasiveness, metastasis and the production of transforming growth factor (TGF)-β, ultimately driving fibrosis [79]. 

Chimeric antigen receptors (also known as zetakines) have been constructed using IL-13 muteins (mutated cytokines) that bind preferentially to IL13Rα2. Initial efforts were undertaken with a mutein that binds selectively (but not exclusively) to IL13Rα2 over IL13Rα1 [80]. Safety concerns regarding the systemic use of such CAR T-cells were fuelled by the widespread expression of IL13Rα1 in normal tissues [77]. To mitigate risk, regional delivery strategies have been undertaken. A first-in-man pilot study was undertaken in which a first-generation IL-13 zetakine CAR was introduced into CD8^+^ cytotoxic T-cell clones from patients with surgically excised GBM [81]. One expanded clone was repeatedly administered up to 12 times to the tumour resection cavity following surgical clearance at the time of the first relapse. Treatment was generally well tolerated, although neurotoxicity was noted in one patient. Brain inflammation was observed in all subjects, as indicated by increased gadolinium enhancement on magnetic resonance imaging and an increased signal on FLAIR images. Pointers towards efficacy included evidence of target antigen downregulation, tumour necrosis and lack of recurrence at the infusion site in some subjects. Allogeneic IL-13 zetakine CAR T-cells also have been engineered using zinc finger nucleases to knock out the glucocorticoid receptor, enabling treatment with dexamethasone to attenuate inflammation following intracerebral CAR T-cell delivery and delay immune rejection of the cells [82]. Administration of four doses of this oligoclonal (non-alloreactive) cell product to the tumour site was well tolerated, without graft versus host disease and with evidence of tumour necrosis in some cases.

In parallel, second-generation CARs have been developed in which a 4-1BB co-stimulatory domain and a mutated IgG4 Fc linker were added, thereby obviating Fc receptor interactions. Cell products were manufactured using central memory-enriched T-cells (e.g., CD45RA−CD62L+). One patient with multifocal leptomeningeal disease characterised by heterogenous IL13Rα2 expression underwent resection of three of five tumours, after which six doses of CAR T-cells were administered using the intracavitary route each week [83]. However, two new lesions appeared at distant sites, prompting a change in the route of administration to the intraventricular route. Ten additional treatments were administered in this manner at 1–3-week intervals. Reduction in tumour size was progressive over the infusion period, resulting in a complete remission followed by relapse with new tumours detected 228 days after first CAR T-cell treatment. Treatment was generally well tolerated, with no toxicity of grade 3 or greater. A more recent update indicates that there has been a second CR in this clinical trial, that the number of intratumoural T-cells prior to treatment was linked to prolonged survival, and that there are plans to combine this approach with an oncolytic viral therapy [84].

Clinical trials of IL13Rα2-targeted CAR T-cells known to be currently recruiting are listed in Table 8.

## 8. Prostate-Specific Membrane Antigen (PSMA)

Prostate-specific membrane antigen (PSMA) is expressed by the majority of prostate carcinomas, including metastatic castrate-resistant disease (mCRPC). Some normal tissues also express low levels of this antigen, including proximal renal tubule cells, salivary glands, bladder, prostate, liver, oesophagus, stomach, small intestine, colon, breast, adrenal gland, testis, ovary and fallopian tubes [85]. Expression has also been detected in the brain, including hippocampal neurons and type 2 astrocytes [85,86]. In addition to prostate tumours, aberrant expression of PSMA has also been reported in the neovasculature associated with multiple solid tumour types [87]. Moreover, PSMA is upregulated in the neovasculature and associated with a number of non-malignant CNS pathologies, including cerebral ischaemic stroke [88,89] or inflammatory processes such as cerebral tuberculosis [90] or neurocysticercosis [91].

Extensive pre-clinical investigation of CAR T-cells directed against PSMA has been undertaken, providing a rationale for the clinical advancement of this target. The first such report was by Junghans et al., employing a first-generation design in which CD3ζ alone provided signalling [92]. Patients with prostate cancer were conditioned with fludarabine and cyclophosphamide and received either 10^9^ or 10^10^ CAR T-cells, followed by low-dose IL-2, administered by continuous intravenous infusion. Treatment was well tolerated with expected cytopenias, but no evidence of on-target off-tumour toxicity. Two patients had a drop in serum prostate-specific antigen (PSA) levels. In parallel with this study, Slovin et al. tested a second-generation CD28-containing CAR [93], administered at a dose of 1–3 × 10^7^ CAR T-cells/kg after a less intensive conditioning regimen [94]. Disease stabilisation was the best result achieved, with evidence of tolerable CRS.

More recently, an armoured CAR T-cell approach has been developed in which a dominant-negative receptor for TGF-β (dnTGF-βR) was co-expressed with a 4-1BB-containing PSMA-specific CAR [95]. This approach set out to neutralise the prototypic immunosuppressive cytokine, TGF-β, although pre-clinical evidence had indicated a possible risk of autoimmunity [96] and, in the transgenic setting, lymphoproliferation [97]. Non-clinical studies confirmed that inclusion of the dnTGF-βR led to a clear enhancement of anti-tumour activity, accompanied by increased expansion of circulating CAR T-cells and the early onset of xenogeneic graft versus host disease (GvHD) in treated mice [95]. Based on these data, a first-in- man clinical trial involving 13 treated patients with castrate-resistant prostate cancer was undertaken (NCT03089203) [98]. One patient developed grade 3 encephalopathy, while a second lymphodepleted subject exhibited clonal CAR T-cell expansion accompanied by a 98% reduction in PSA but was followed by death due to CRS (accompanied by hyper-ferritinaemia) and sepsis. Following this event, dose de-escalation to 1–3 × 10^7^ cells/m^2^ was undertaken, administered after lymphodepletion with cyclophosphamide and fludarabine. Treatment was generally well tolerated thereafter. A 30% reduction in serum PSA of limited duration was noted in two of these patients, and CAR T-cells were detected in 7 of 9 metastatic tumour biopsies, collected at or close to day 10. The best radiographic response was stable disease.

Results of this trial informed the design of a second study, sponsored by Tmunity Therapeutics [99]. However, toxicity leading to two patient deaths resulted in the premature closure of this study. One patient developed severe immune effector cell-associated neurotoxicity syndrome (iCANS), accompanied by macrophage activation syndrome, while a second grade 2 iCANS event was also noted. Investigators then added in prophylactic anakinra together with a CAR T-cell dose reduction. However, a second patient death occurred in which macrophage activation may also have been responsible, as indicated by profound hyper-ferritinaemia. Decreases in PSA levels were noted in 4 of 7 patients. In light of this toxicity, Tmunity have re-designed the CAR, substituting the 4-1BB domain with a CD2 module [100]. In addition to dnTGF-βR, T-cells also co-express a PD1-CD28 switch receptor. Using this tricistronic construct, anti-tumour immunity is maintained with lowered cytokine release. Clinical testing is currently ongoing (NCT05489991).

Autologous CD28-containing PSMA-specific CAR T-cells are also under development by Poseida Therapeutics [101]. In contrast to the aforementioned studies, in which integrating viral vectors are used to achieve gene transfer, Poseida are using the PiggyBac transposon system. Use of this vector system is compatible with much larger genetic cargo sizes and the attainment of cell products in which less differentiated stem cell memory T-cells are enriched. Cell products are infused after lymphodepletion with cyclophosphamide and fludarabine. The trial was placed on clinical hold after one early patient died of liver failure in association with features of macrophage activation syndrome and uveitis (https://www.sec.gov/Archives/edgar/data/1661460/000119312520222442/d948162d8k.htm, accessed on 31 October 2022). This prompted a decision to dose de-escalate to the -1 dose level. Thereafter, however, results have been encouraging, with a drop in PSA seen in 7 patients and favourable responses seen using PSMA positron emission tomography scans in 3 patients.

In summary, CAR-mediated targeting of PSMA has resulted in a reduction in tumour burden in a number of cases. However, inflammatory toxicity has proven challenging with this target, particularly when measures designed to enhance T-cell expansion and persistence are employed (e.g., dnTGF-βR). The relationship between CNS toxicity and PSMA expression at that location also warrants further study. Clinical trials of PSMA-targeted CAR T-cells known to be currently recruiting are listed in Table 9.

## 9. GD2 (Disialoganglioside 2)

Disialoganglioside D2 (GD2) is expressed by virtually all neuroblastomas in addition to diffuse midline gliomas, melanomas and sarcomas. A number of early-phase clinical trials in neuroblastoma patients have provided encouragement for CAR T-cell targeting of GD2 in this disease. A study with a first-generation CAR reported that 3 of 11 children with neuroblastoma achieved CR after infusion of these cells, in the absence of dose-limiting toxicities [102,103]. Immunotherapy using second-generation GD2 CAR T-cells also proved to be safe, with evidence of tumour shrinkage noted in some cases [104]. However, third-generation CAR T-cells have yielded less impressive results, with stable disease reported as the best response in two separate studies [105,106]. An additional trial of a third-generation CAR targeted against GD2 and with a distinctive optimised design is currently ongoing in Rome [107].

In addition to conventional T-cells, GD2 CAR-expressing invariant NKT-cells are under study in children with relapsed neuroblastoma [108]. Cells are armoured with IL-15 to enhance persistence. Of the 11 treated patients, there was 1 CR, 1 PR and 4 patients who achieved stable disease. Treatment was well tolerated.

GD2 is also expressed in diffuse midline gliomas, which are incurable with all current therapies. A recent study in 4 patients demonstrated the feasibility of treatment with GD2-targeted CAR T-cells in this very difficult-to-treat patient population, with the implementation of measures to mitigate the risks of inflammation at this critical CNS location [109]. CAR T-cells were initially administered intravenously, followed by subsequent intraventricular dosing via an Ommaya reservoir. Three patients exhibited clinical and radiographic tumour responses. Clinical trials of GD2-targeted CAR T-cells known to be currently recruiting are listed in Table 10.

## 10. B7 Family Members

### 10.1. B7-H3 (CD276)

B7-H3 is expressed on over 60% of tumours including neuroblastoma, pancreatic, ovarian, breast, prostate, lung, stomach, squamous cell and clear cell renal cell carcinomas, glioblastoma, paediatric brain tumours and sarcomas [110]. Importantly, B7-H3 is found on cancer initiating cells, tumour-associated vessels and stromal fibroblasts. Expression has been linked to enhanced invasiveness and worsened prognosis. Although B7-H3-encoding mRNA is widely expressed in normal tissues, B7-H3 protein is expressed at limited levels in normal tissues, including activated dendritic cells, monocytes, lymphocytes, spermatocytes and the pituitary gland.

Clinical experience with B7-H3-targeted CAR T-cells is expanding. One patient with glioblastoma received repeated dosing of CAR T-cells via the intracavitary route using an Ommaya reservoir [111]. A PR ensued that lasted 50 days. Headache was the main form of toxicity. Relapse was assumed to result from antigen loss owing to the heterogeneity of B7-H3 expression in the tumour. Similarly, a partial response was reported in a patient with multiple basal cell carcinomas following repeated intra-tumoural delivery of the CAR T-cells [112]. Although liver enzyme abnormalities were noted post-treatment, no serious adverse events occurred. The first two subjects with diffuse intrinsic pontine glioma who were treated in the BrainChild-03 clinical trial both received repeated intraventricular CAR T-cell doses [113]. Grade 2 fever and headache occurred after infusions, and a transient episode of focal weakness and dysarthria occurred after infusion number 8 in one patient. Stable disease was achieved in both subjects, lasting for at least 138 days in one case. The STRIVE-02 study is a first-in-human trial in children and young adults with B7-H3-expressing solid tumours [114]. Nine subjects have been treated to date, with stable disease seen in three cases. These include a partial metabolic response, as detected by FDG-PET scanning. One case of transient dose-limiting grade 4 liver enzyme elevation was reported. A patient with an anaplastic meningioma has also received B7-H3 targeted CAR T-cells using an Ommaya device, and evidence of local tumour control was observed with acceptable safety [115].

A large number of clinical trials involving B7-H3-targeted CAR technologies are currently ongoing, as summarised in Table 11.

### 10.2. Programmed Death Receptor Ligand 1 (PD-L1)

Programmed death receptor ligand 1 is widely expressed on a range of solid tumours and has been successfully targeted using CAR T-cells in a number of pre-clinical studies [116,117]. A further theoretical attraction of this approach is the demonstration that PD-L1-targeted CAR T-cells can promote the upregulation of this target in PD-L1 low tumour cells [118]. Balancing this, it was also noted that amplified PD-L1 expression in normal tissues could have detrimental effects due to impairment of PD-L1-dependent tolerance mechanisms in vital organs. Natural killer cells have also been engineered to express a PD-L1 specific CAR and effectively killed a broad range of tumour cell lines [119] while also eliminating immunosuppressive myeloid cells that express high levels of PD-L1 [120]. However, a CAR T-cell clinical trial in NSCLC was terminated in China owing to an undisclosed serious adverse event (NCT03330834, accessed on 7 January 2023) which responded to treatment with the anti-IL-6 receptor antibody, tocilizumab and steroids [118]. One other clinical study has safely employed a PD1/CD28 switch receptor, which was co-expressed with a CD19-specific 4-1BB-containing CAR [121].

## 11. Glypican 3

Glypican-3 is an oncofoetal heparan sulphate proteoglycan. It is expressed in more than 70% of hepatocellular carcinomas (HCC) in addition to some ovarian clear cell carcinomas, squamous cell lung carcinomas, melanomas, hepatoblastomas, Wilms’ tumours and α foetoprotein-producing gastric carcinomas. By contrast, minimal expression in normal tissue has been reported, with a predominantly cytoplasmic distribution [122].

Glypican-3-specific CAR T-cell immunotherapy has been evaluated in 13 patients with HCC. Two partial responses were achieved in this cohort, although one patient succumbed due to CRS [123]. Subsequently, an additional HCC patient received repeated dosing with the same CAR T-cell product in combination with a multi-kinase inhibitor, sorafenib. Combination therapy was well tolerated and led to a sustained complete response [124]. In one small study, Glypican-3-specific CAR T-cells were armoured to co-express IL-7 and CCL19 [18]. These were administered using the intra-tumoural route to three metastatic deposits in a patient with hepatocellular carcinoma. One of these lesions, which was located in the liver, regressed completely without adverse effects, giving an overall PR by RECIST criteria. Alternatively, glypican-3-specific CAR T-cells have been armoured with an unnamed transcription factor, leading to one PR in six treated patients with manageable safety [125]. Clinical trials of Glypican-3 targeted CAR T-cells known to be currently recruiting are listed in Table 12.

## 12. NKG2D Ligands

NKG2D ligands are expressed on 80% of human cancers and are recognised by natural killer (NK) and some T-cell subsets, enabling the elimination of transformed and otherwise potentially dangerous cells. To potentiate this physiological immune surveillance system, it was logical that CARs targeted against NKG2D ligands would be developed [126]. Ample clinical experience that supports the safety and potential efficacy of this strategy has been generated by Celyad Oncology employing a CAR in which CD3ζ was fused to the endodomain of NKG2D itself. Over 100 patients have been safely treated using this approach in a series of clinical trials, which are summarised below. Although many treated patients had haematological malignancies, these are nonetheless informative with respect to the potential safety of this group of 8 target ligands.

Initial studies were conducted in the absence of lymphodepletion. The first human trial was a 3+3 dose-escalation study in acute myeloid leukaemia (AML), myelodysplastic syndrome (MDS) and myeloma [127]. Doses were modest, ranging from 1 × 10^6^ to 3 × 10^7^ cells. There were no DLTs in the 12 treated patients. One patient had haematological improvement for 3 months at the highest dose level. Notably, concomitant viral (influenza or parainfluenza) or bacterial infection (Gram-negative biliary sepsis) was not accompanied by significant inflammatory or autoimmune toxicity, despite the risk of upregulation of NKG2D ligands under these circumstances.

In the THINK study [128,129], NKG2D-targeted CAR T-cells were administered at doses of 3 × 10^8^, 1 × 10^9^ or 3 × 10^9^ cells per infusion to patients with AML, MDS or multiple myeloma without lymphodepletion. A total of 25 patients were recruited during the dose-escalation phase. Three CAR T-cell doses were administered every 1–2 weeks, with the possibility to consolidate with three further doses in the absence of progressive disease. During dose escalation, 3 AML patients achieved CR (1 with an incomplete recovery of counts), although durability was less than 3 months [130]. The 3 × 10^8^ dose level was expanded in a cohort of 8 AML patients. Stable disease was the best response observed. Cytokine release syndrome occurred in 13 patients (4 grade 3, 2 grade 4) and was responsive to tocilizumab in all cases. One case of grade 3 ICANS was also seen (dose level unclear). Furthermore, one grade 4 pneumonitis event occurred in a patient treated at the lowest dose level, although this was not encountered at either of the higher dose levels.

The THINK study also included a solid tumour arm [131], in which CAR T-cells were administered without conditioning at doses of 3 × 10^8^ (n = 4), 1 × 10^9^ (n = 4), or 3 × 10^9^ (n = 6) cells to patients with colorectal, ovarian or pancreatic cancer. One DLT occurred at the highest dose level, presenting as grade 4 CRS. Stable disease was seen in 4 patients.

The THINK study was amended to incorporate conditioning with cyclophosphamide 300 mg/m^2^ and fludarabine 30 mg/m^2^ each for 3 days prior to 3 × 10^8^ CAR T-cells. Although chemotherapy may potentially lead to NKG2D ligand upregulation in an unpredictable manner, no DLTs were reported in 3 patients with metastatic colorectal cancer. One patient achieved stable disease.

In the SHRINK study, concurrent FOLFOX (folinic acid, 5-fluorouracil and oxaliplatin) was administered in conjunction with CAR T-cells, once again imposing a risk of unpredictable NKG2D ligand upregulation. Three cycles of this combination were administered every 2 weeks in a 3+3 design that incorporated three dose levels of 1 × 10^8^, 3 × 10^8^ and 1 × 10^9^ cells (e.g., nine patients in total). Colorectal cancer patients were treated either in the neoadjuvant setting (n = 4) or with refractory disease (n = 5). CAR T-cells were administered 48 h after the completion of the chemotherapy cycle, with the potential for further cycles in the absence of progressive disease. There were no DLTs in the dose escalation phase of the study. One patient in the neoadjuvant cohort achieved a PR, while two achieved stable disease. In the refractory cohort, 4 of 5 patients achieved stable disease [132].

In the DEPLETHINK study [129,133], NKG2D-targeted CAR T-cells were administered to 17 patients with AML and MDS after lymphodepleting chemotherapy with fludarabine and cyclophosphamide. Three CAR T-cell dose levels were evaluated in the induction phase, namely 1 × 10^8^, 3 × 10^8^ and 1 × 10^9^ cells, each administered as a single dose. A potential CAR T-cell consolidation cycle was included on days 36, 50 and 64. Only grade 1 or grade 2 toxicities were seen (CRS and diarrhoea). However, grade 4 CRS and grade 3 neurotoxicity were seen in one of these patients following a second infusion of 3 × 10^9^ cells as consolidation (without prior lymphodepletion). Better engraftment was seen with a 3-day rather than 7-day interval between lymphodepletion and CAR T-cell infusion. However, no objective responses were seen.

The ALLOSHRINK study [134] employed a starting dose of 1 × 10^8^ allogeneic CAR T-cells, administered to patients with metastatic colorectal cancer. Cells incorporated an endodomain-truncated CD3ζ mutant in order to disable the endogenous T-cell receptor/CD3 complex. Three doses of this product (e.g., total dose of 3 × 10^8^ NKG2DL-targeted CAR T-cells) were administered IV at 2 weekly intervals. CAR T-cells were co-administered with FOLFOX. Dose escalation proceeded to 3 × 10^8^ and 1 × 10^9^ cells in a 3+3 design, meaning that nine patients received the highest dose. In the first 15 patients treated in the ALLOSHRINK trial, 2 PRs were reported, while 9 patients achieved stable disease without any DLTs. However, when the ALLOSHRINK treatment combination was escalated to a dose of 1 × 10^9^ cells per cycle, administered on 3 separate cycles, together with FOLFOX and pembrolizumab (KEYNOTE-B79), two fatalities were reported in which similar pulmonary findings were noted, ultimately leading to the suspension of the programme. In considering the potential underlying cause of these toxic events, it should be noted that similar events were not observed prior to the addition of pembrolizumab to the therapeutic regimen. Indeed, in one report, it is stated that twenty-five patients were treated with CYAD-101 in the ALLOSHRINK trial, without any DLTs (https://www.onclive.com/view/fda-places-clinical-hold-on-keynote-b79-trial-examining-car-t-cell-therapy-cyad-101-in-metastatic-crc, accessed on 14 December 2022). A second potentially important point is that the allogeneic T-cells used in this study did retain some endogenous T-cell receptor activity [135]. This raises the possibility that alloreactivity was a co-factor in these events, further unmasked by the introduction of PD1 blockade [136].

NKG2D ligands are also being targeted by Nkarta using allogeneic CAR-engineered NK cells, administered to patients with AML (https://ir.nkartatx.com/news-releases/news-release-details/nkarta-announces-positive-preliminary-dose-finding-data-two-lead, accessed on 14 December 2022). Three of five patients who received the higher dose level of 1–1.5 billion CAR NK cells per dose achieved a complete response with full haematological recovery. In two of these cases, the CR was negative for minimal residual disease. Impressively, there were no DLTs or cases of CRS, ICANs or GvHD.

Clinical trials of NKG2D-targeted CAR technologies known to be currently recruiting in the solid tumour arena are listed in Table 13.

## 13. Prostate Stem Cell Antigen (PSCA)

PSCA is primarily expressed in the prostate gland but is also found in the kidney, pancreas, bladder and central nervous system [137]. Expression is upregulated in tumours derived from these tissues. A phase 1 clinical trial of PSCA-specific CAR T-cells is ongoing at the City of Hope Medical Center in patients with metastatic prostate cancer. Twelve patients have been treated to date, and two DLTs due to cystitis were identified in patients who received cyclophosphamide and fludarabine-based lymphodepletion. This prompted a reduction in the cyclophosphamide dose, after which no DLTs occurred in 3 further subjects. Although stable disease was the best response seen, a drop in PSA of 90% was observed in one subject [138]. A second study sponsored by Bellicum Pharmaceuticals incorporates their GoCAR-T technology, in which a first-generation CAR is co-expressed with a rimiducid-activated co-stimulatory module derived from MyD88 and CD40. One patient with pancreatic cancer died following treatment with this product. Although reported as unrelated to CAR T-cell treatment, this event prompted a temporary hold of the study, which has now been reopened. Subsequently, a partial response was reported in one patient with prostate cancer in late 2021 (https://ir.bellicum.com/news-releases/news-release-details/bellicum-announces-positive-interim-data-phase-12-gocar-tr, accessed on 14 December 2022). Details of both of these ongoing studies are summarised in Table 14.

## 14. Carcinoembryonic Antigen (CEA)

Carcinoembryonic antigen is naturally expressed by the luminal surface of the gastrointestinal and respiratory tract, where theoretically it should not be accessible to CAR T-cells (akin to MUC1). Upregulated expression of non-polarised CEA is a feature of many cancers, most notably colorectal carcinoma. However, treatment of lymphodepleted colorectal cancer patients with first-generation CEA-targeted T-cells plus IL-2 did not achieve any objective responses and was marred by the poor persistence of the CAR T-cells in vivo [139]. Acute respiratory toxicity was noted in the highest dose cohort, prompting the early closure of the trial. A dose-escalation study involving a second-generation CAR did not uncover serious adverse effects, although stable disease was the best response seen [140]. Nonetheless, some tumour shrinkage was noted in two patients in this study. Both studies also reported a drop in circulating CEA levels in some patients. In a further study, CEA-specific CAR T-cells were infused via the hepatic artery in patients with CEA+ liver metastases [141,142]. Although well tolerated, the best response was stable disease, although in one case, a complete metabolic response was documented by FDG-PET scanning.

Clinical trials of CEA-targeted CAR technologies known to be currently recruiting are listed in Table 15.

## 15. CD70

CD70 is expressed on multiple haematological malignancies in addition to a number of solid tumours, most notably clear cell renal cell carcinoma. Lower levels of CD70 expression have been reported on a variety of other solid tumour types, including pancreatic cancer, melanoma, ovarian, lung, colonic and head and neck carcinomas, mesothelioma and glioblastoma. Expression tends to be higher in metastatic lesions, and it is linked to an adverse prognosis. CD70 is also expressed on stromal cells found in tumours such as cancer-associated fibroblasts and endothelial cells. Although clinical data in the solid tumour setting are limited, CRISPR Therapeutics recently reported that of 13 evaluable patients in its phase 1 COBALT-RCC trial (renal cell carcinoma), 1 patient achieved a PR that deepened to become a CR which was maintained at 18 months [143]. The safety profile was acceptable. Clinical trials of CD70-targeted CAR technologies known to be currently recruiting in the solid tumour arena are listed in Table 16.

## 16. Carboxy Anhydrase IX (CAIX)

Carboxy anhydrase IX is expressed in renal cell carcinoma and was one of the early targets evaluated clinically using first-generation CAR T-cells. Among the first three treated patients, two developed on-target off-tumour toxicity due to unanticipated biliary expression of CAIX [144]. Investigators found that this toxicity could be ameliorated by pre-treatment with a blocking antibody directed against CAIX [145]. However, no clinical responses were observed in this clinical trial. There is a single ongoing study of CAIX-targeted CAR T-cells in renal cell carcinoma sponsored by the Affiliated Hospital of Xuzhou University (NCT04969354).

## 17. CD133

CD133 is a stem cell marker that is expressed on a range of solid tumours, including hepatocellular carcinoma, glioblastoma, pancreatic cancer, gastric cancer, colorectal cancer and endothelial cells implicated in neovascularization. Although it is expressed on normal haematopoietic and other stem cells, levels are lower than on transformed cells. Phase 1 testing of CD133-targeted CAR T-cells was conducted in 23 patients with hepatocellular, pancreatic and colorectal cancer. Three PRs were achieved, and the main toxicities observed were transient cytopaenias and hyperbilirubinaemia (due to the worsening of pre-existing biliary tract obstruction) [146]. A case report described a patient with cholangiocarcinoma who was treated with successive infusions of EGFR- and CD133-specific CAR T-cells [147]. The former resulted in an 8.5-month PR, while the latter achieved a 4.5-month PR. Grade 3 cutaneous and mucosal toxicity was attributed to the CD133-specific cells. A phase II clinical trial of CD133-specific CAR T-cells was conducted in patients with hepatocellular carcinoma, a disease in which CD133 is associated with a poorer outcome [148]. One of 21 patients achieved a PR. Once again, hyperbilirubinaemia due to aggravated biliary tract obstruction was the main toxicity noted. Clinical trials of CD133-targeted CAR technologies known to be currently recruiting in the solid tumour arena are listed in Table 17.

## 18. Erythropoietin-Producing Human Hepatocellular Carcinoma (Ephrin) Type A Receptor 2 (EphA2)

Ephrin type A receptor 2 (EphA2) is expressed in paediatric sarcomas and paediatric high-grade gliomas (HGGs), glioblastoma multiforme, breast cancer, NSCLC and oesophageal cancer. It has been linked to a worsened prognosis in HGGs [149] and other tumours. It is also expressed on cancer-associated fibroblasts, providing a stromal target for CAR T-cell attack. However, Protein Atlas reveals that mRNA encoding EphA2 is highly expressed in the proximal digestive tract (https://www.proteinatlas.org/ENSG00000142627-EPHA2/tissue, accessed on 12 January 2023), while expression in the lung has also been reported [150]. In a preliminary report of intravenous CAR T-cell immunotherapy of recurrent glioblastoma (NCT03423992, sponsored by Xuanwu Hospital, Beijing), one of three treated patients achieved disease stabilisation [150]. However, two of the patients developed pulmonary oedema (responsive to dexamethasone), raising a question about on-target off-tumour toxicity. No other CAR T-cell trials directed against this target are listed as actively recruiting on clinicaltrials.gov (accessed on 18 December 2022). Notably, a clinical trial involving an EphA2-targeted antibody–drug conjugate was terminated due to bleeding and coagulation-related toxicity [151]. Although well tolerated, the development of a humanised IgG1 antibody directed against EphA2 was recently halted due to poor tumour uptake and efficacy [152]. On the other hand, Bicycle Therapeutics are developing a peptide toxin conjugate targeted against EphA2 and recently announced acceptable toxicity of this agent and no evidence of coagulopathy (https://www.businesswire.com/news/home/20220907005284/en/Bicycle-Therapeutics-Announces-BT5528-Phase-I-Dose-Escalation-Results-in-Patients-with-Advanced-Solid-Tumors, accessed on 18 December 2022). They also presented preliminary evidence of the anti-tumour activity of this agent, including one CR and three PRs among forty-five treated patients.

## 19. Fibroblast Activation Protein (FAP)

Fibroblast activation protein is expressed by cancer-associated stroma in virtually all epithelial cancers. It is also expressed during wound healing and in chronic inflammatory and fibrotic conditions but is absent in most quiescent adult stromal cells. Complete ablation of FAP-expressing cells in transgenic mice caused anaemia and cachexia due to loss of fibroblasts in bone marrow and muscle [153]. However, targeting of FAP in immune competent mice was safe and resulted in inhibition of tumour growth [154]. A Phase I clinical trial of FAP-targeted CAR T-cells has been undertaken in patients with malignant pleural mesothelioma [155]. Owing to the propensity of this tumour to undergo local intracavitary spread, cells were infused into the pleural cavity. Safety was good, although a dose of only 1 million CAR T-cells was infused in 3 subjects. The impact on patient outcome was not evaluated. There are no actively recruiting clinical trials of FAP-targeted CAR T-cells listed on clinicaltrials.gov (accessed on 6 January 2023).

## 20. Adhesion Molecules

### 20.1. Epithelial Cell Adhesion Molecule (EpCAM)

Epithelial cell adhesion molecule (EpCAM) exhibits polarised expression on normal epithelial cells, whereas this polarity is lost upon malignant transformation. Several clinical trials involving EpCAM-specific CAR T-cells are ongoing. An early phase clinical trial of EpCAM-specific CAR T-cells in colorectal and gastric cancer patients reported no dose-limiting toxicities in the first five treated subjects, although one patient developed immune hepatitis, which prolonged hospitalisation [156]. Stable disease was reported in 4 of 5 cases. One case report described a patient with malignant mesothelioma who developed severe CRS accompanied by pulmonary oedema that was ultimately controlled using tocilizumab and corticosteroids [157]. Notably, murine pre-clinical studies have demonstrated the potential for EpCAM-targeted CAR T-cells to cause pulmonary immunopathology [158]. Clinical trials of EpCAM-targeted CAR technologies known to be currently recruiting in the solid tumour arena are listed in Table 18.

### 20.2. Neuronal L1 Cell Adhesion Molecule (L1CAM; CD171)

Neuronal L1 cell adhesion molecule (L1CAM; CD171) is expressed in virtually all neuroblastomas in addition to neuroendocrine tumours, small cell lung carcinoma, melanoma, glioblastoma multiforme, sarcomas, ovarian serous cancer, colon cancer, renal cell carcinoma, prostate cancer and malignant mesothelioma. It is also expressed in peripheral nerves, the brain, vascular endothelial organs, skin and renal collecting ducts (https://www.proteinatlas.org/ENSG00000198910-L1CAM/tissue, accessed on 7 January 2023). A clinical trial was undertaken several years ago in patients with neuroblastoma who received one or more infusions of L1CAM-retargeted CAR T-cell clones [159]. Treatment was well tolerated without evidence of on-target off-tumour toxicity. One of six treated patients achieved stable disease and survived for four and a half years.

## 21. Roundabout Guidance Receptor 1 (ROBO1)

Roundabout guidance receptor 1 (ROBO1) plays an important role in neurodevelopment and is aberrantly expressed in glioma and pancreatic ductal adenocarcinoma (PDAC). A case report described a PDAC patient who received irradiated NK92 cells engineered to express a ROBO1-specific CAR [160]. Repeated infusions were administered using the intravenous and intra-tumoural routes (to liver metastases) (NCT03941457).

Although disease was stable over 5 months, tumour regression was not observed. Although Protein Atlas indicates that ROBO1 is expressed in the brain and at lower levels in other organs (https://www.proteinatlas.org/ENSG00000169855-ROBO1/tissue, accessed on 10 January 2023), infusions were well tolerated. No active CAR clinical trials directed against this target were identified on clinicatrials.gov (accessed on 10 January 2023).

## 22. Conclusions

In this review, we have surveyed the clinical experience of CAR-based immunotherapy in the solid tumour setting, focusing in particular on target selection and clinical outcome. It is clearly evident that efficacy does not mirror that seen in B-cell and plasma cell malignancies, reflecting the multiple additional challenges that hinder solid tumour CAR immunotherapy. As indicated above, on-target off-tumour toxicity is a major concern since no target is absolutely tumour specific. Transient expression systems can be used to initially de-risk a novel target, but these are not absolutely predictive of safety in subsequent studies that have been performed using stable gene transfer systems. Efficacy has been disappointing on the whole, although there are pointers towards greater efficacy when more tumour-selective targets are selected (e.g., claudin 18.2, claudin 6). This may reflect the fact that low-level expression of target antigen in normal tissue may not only increase the risk of toxicity but may also compromise efficacy through retention of CAR T-cells at extra-tumoural locations (e.g., large antigen sink effect). Repeated dosing has been linked to improved efficacy in some cases, with examples illustrating how responses can deepen with additional dose administration. Similarly, regional delivery in selected tumour types can achieve improved delivery to the site of disease, once again improving both safety and efficacy profiles. In tumour types where regional delivery in not a feasible strategy, armouring of CAR T-cells with an appropriate chemokine (e.g., CCL19) or chemokine receptor can increase efficiency of tumour delivery, an approach that has boosted both efficacy and safety in pre-clinical testing [161]. Polarised expression of targets in normal epithelia has been suggested to enhance safety since CAR T-cells should theoretically not have access to the luminal surface of such tissues. Nonetheless, the demonstration of significant pulmonary toxicity in a trial of CEA-targeted CAR T-cells suggests that this may only afford limited protection against on-target off-tumour toxicity. Post-translational modifications of some cell surface molecules that are radically different in transformed compared to normal tissue (e.g., tumour-associated glycoforms of MUC1) may offer an interesting alternative direction to the identification of safer targets for solid tumour CAR T-cell immunotherapy. In addition, logic-gated strategies that couple maximum CAR T-cell activation to the recognition of two or more tumour-associated target antigens [162] may assist in improved CAR T-cell discrimination between healthy and diseased tissue, although this approach remains to be tested in the clinic in the context of solid tumours. On a cautionary note, pre-clinical studies have suggested that dual-targeted CAR T-cells may acquire greater sensitivity to recognise very low levels of target antigen due to the “docking effect” provided by the second chimeric receptor [163]. Finally, the hostile nature of the tumour microenvironment remains a formidable challenge, and there is emerging evidence that we will also need to deal with an immunosuppressive, inflammatory “counterattack” by the tumour as well as issues seen in the context of blood cancers such as antigen loss. On the positive side, however, we have seen clinical evidence that epitope spreading may ensue following clinical CAR T-cell immunotherapy, providing additional opportunities to boost efficacy where it is needed most.

## Figures and Tables

**Figure 1 biology-12-00287-f001:**
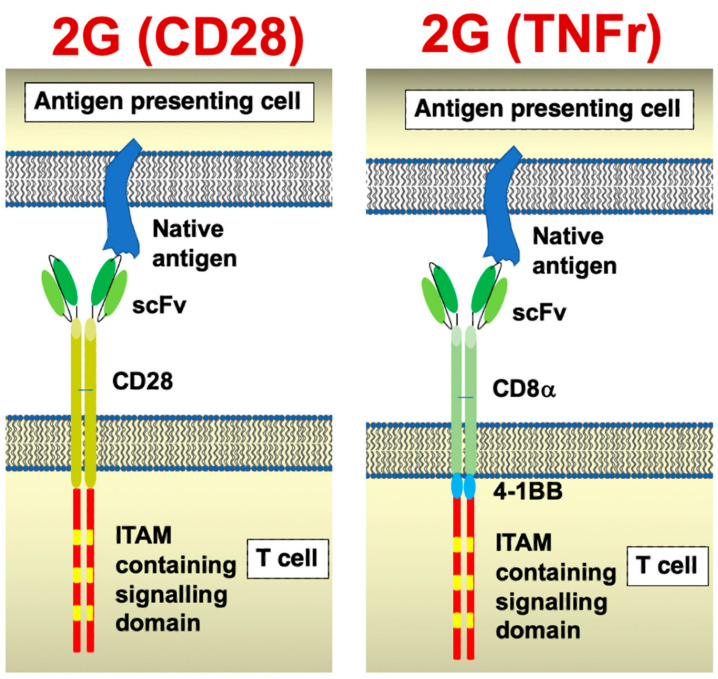
Chimeric antigen receptor structure. Cartoons illustrate the structures of a CD28 and a tumour necrosis factor receptor (TNFr)-containing second-generation CARs. Most commonly, 4-1BB is incorporated into the latter CAR platform.

**Figure 2 biology-12-00287-f002:**
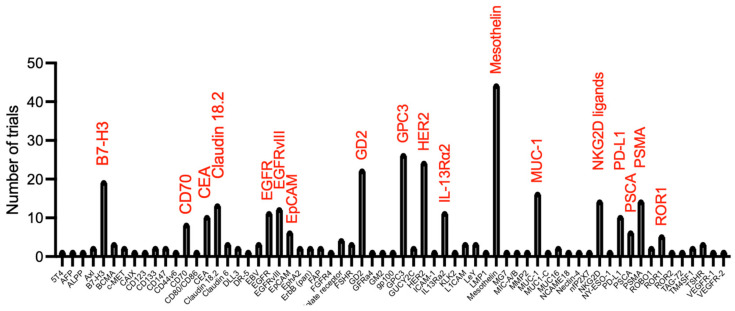
Target selection in currently registered CAR T-cell clinical trials in solid tumour indications (analysis on Beacon database undertaken 7 November 2022).

**Table 1 biology-12-00287-t001:** Ongoing CAR T-cell clinical trials directed against mesothelin (assessed 15 December 2022).

Disease	Sponsor	Notes	Identifier
Mesothelin+ tumours	Ruijin Hospital	mRNA electroporated CAR T-cells	NCT04981691
Mesothelin+ tumours	2nd Affiliated Hospital Guangzhou Medical University	One of many targets under study	NCT03198052
Mesothelin+ tumours	Shanghai Pudong Hospital		NCT05531708
Ovarian cancer	Weijia Fang		NCT05372692
Pancreatic cancer	Shenzhen BinDeBio Ltd.		NCT03638193
Mesothelin+ tumours	University of Pennsylvania	Investigating different routes of administration	NCT03054298
Gastric cancer	Shenzhen BinDeBio Ltd.	One of multiple targets and indications	NCT03941626
Mesothelin+ tumours	Shanghai Mengchao Cancer Hospital	CAR T-cells secrete PD1 nanobodies	NCT05373147
Mesothelin+ tumours	TCR2 Therapeutics	See text for description of CAR; Also incorporates a PD1/CD28 switch receptor	NCT05451849
Ovarian cancer	Shanghai 6th People’s Hospital		NCT03814447
Mesothelin+ tumours	TCR2 Therapeutics	See text for description of CAR	NCT03907852
Mesothelin+ tumours	1st Affil. Hospital with Nanjing Medical University	Cells co-express CD19 CAR and truncated EGF receptor safety switch	NCT05166070
Pancreatic cancer	1st Affil. Hospital of Harbin Medical University	Explores influence of gut microbiome on CAR T-cell metabolism and function	NCT04203459

**Table 2 biology-12-00287-t002:** Ongoing CAR-based clinical trials directed against HER2 (assessed 29 October 2022).

Disease	Sponsor	Notes	Identifier
CNS tumours	Baylor College of Medicine	Intracranial	NCT02442297
Lung cancer ^1^	Guanghou Med. University	Systemic or regional	NCT3198052
Solid tumours	Bellicum Pharmaceuticals	Dual switch CAR T-cells with rimiducid	NCT04650451
Paediatric CNS tumours	Seattle Children’s Hospital	Intracranial [31]	NCT03500991
HER2 3+ solid tumours	Triumvira	TAC ^2^ comprising HER2-specific DARPin fused to aCD3ε scFv and CD4 spacer, transmembrane and endodomain	NCT04727151
Brain/leptomeningeal malignancy	City of Hope Medical Center	Intracranial	NCT03696030
Breast cancer	Shenzen Geno-Immune Medical Institute	CAR also targets GD2 and CD44v6	NCT04430595
Multiple solid tumours	Baylor College of Medicine	Combination of CAR T-cells with oncolytic adenovirus	NCT03740256
Serosal cavity metastases	Sichuan University	CAR also targets PD-L1	NCT04684459
Multiple solid tumours	Shanghai PerHum Therapeutics	Intravenous delivery	NCT04511871
Brain tumours ^1^	Xuanwu Hospital	Patients may also receive anti-PD-L1	NCT03423992
Sarcoma	Baylor College of Medicine	Patients also receive immune checkpoint blockade	NCT04995003
Ependymoma	Pediatric brain tumor consortium	Intravenous. May be followed by surgery	NCT04903080
Brain tumours	City of Hope Medical Center	Intracranial	NCT03389230
Solid tumours	Carisma Therapeutics	CAR-engineered macrophages administered intravenously	NCT04660929

^1^ HER2 is one of multiple targets; ^2^ T-cell antigen coupler.

**Table 3 biology-12-00287-t003:** Ongoing CAR T-cell clinical trials directed against EGFR (assessed 20 November 2022).

Disease	Sponsor	Notes	Identifier
SCCHN ^1^	King’s College London	Intratumoural delivery of panErbB CAR T-cells	NCT01818323
NSCLC ^2^	Sun Yat-Sen University	CXCR5 armoured CAR T-cells	NCT04153799
NSCLC	2nd Affiliated Hospital Guangzhou Medical University	CXCR5 armoured CAR T-cells	NCT05060796
Solid tumours	Seattle Children’s Hospital	EGFR806 antibody	NCT03618381
Solid tumours	Chinese PLA General Hospital	TGF-β receptor knockout	NCT04976218
Paediatric CNS tumours	Seattle Children’s Hospital	EGFR806 antibody—intracranial delivery	NCT03638167
Solid tumours	2nd Affiliated Hospital Guangzhou Medical University	One of many targets under study	NCT03198052
Lung and TNBC	2nd Affiliated Hospital Guangzhou Medical University	CAR also targets B7-H3	NCT05341492

^1^ Squamous cell carcinoma of head and neck; ^2^ non-small cell lung carcinoma.

**Table 4 biology-12-00287-t004:** Ongoing CAR T-cell clinical trials directed against EGFRvIII (assessed 20 November 2022).

Disease	Sponsor	Notes	Identifier
Glioma	Xuanwu Hospital, Beijing	One of many targets under study	NCT03423992
Glioblastoma	2nd Affiliated Massachusetts General Hospital	CAR T-cells co-express an EGFR-specific T-cell engager	NCT05024175
Multiple solid tumours	Shenzhen BinDeBio	One of many targets under study	NCT03941626

**Table 5 biology-12-00287-t005:** Ongoing CAR T-cell clinical trials directed against MUC1 (assessed 14 December 2022).

Disease	Sponsor	Notes	Identifier
Solid tumours	2nd Affiliated Hospital Guangzhou Medical University	One of multiple targets	NCT03198052
Breast cancer	Minerva Biotechnologies Corp.	Targets MUC1*	NCT04020575
Solid tumours	Poseida Therapeutics	Allogeneic CAR T-cells targeted against MUC1-C	NCT05239143
Sarcomas	Shenzhen Geno-Immune Medical Institute	One of multiple targets	NCT03356782

**Table 6 biology-12-00287-t006:** Ongoing CAR T-cell clinical trials directed against CLDN18 (assessed 23 November 2022).

Disease	Sponsor	Notes	Identifier
Gastric and pancreatic cancer	Changhai Hospital	Collaboration with CARsgen—current trial status unknown	NCT03159819
Solid tumours	CARSgen Pharmaceuticals	Reference [68]	NCT03874897
Lung cancer	2nd Affiliated Hospital Guangzhou Medical University	One of many targets under study	NCT03198052
Multiple solid tumours	Suzhou Immunofoco Biotechnology Co.	One of many targets under study	NCT05472857
Gastric, GOJ ^1^ and PDAC ^2^	Shenzhen Fifth People’s Hospital		NCT05277987
Solid tumours	CARSgen Pharmaceuticals		NCT04404595
Pancreatic and gastric cancers	Shenzhen University General Hospital		NCT05620732
Solid tumours	Shanghai East Hospital	Developed by Nanjing Legend Biotech Co.	NCT04467853
Gastric, GOJ and PDAC	CARsgen Therapeutics Co.		NCT04581473

^1^ Gastro-oesophageal junction; ^2^ pancreatic ductal adenocarcinoma.

**Table 7 biology-12-00287-t007:** Ongoing CAR T-cell clinical trials directed against FRα (assessed 23 November 2022).

Disease	Sponsor	Notes	Identifier
Ovarian, fallopian tube and peritoneal cancer	University of Pennsylvania	Enrolment requires ≥2 + FRα staining in ≥70% of tumour cells. Cells are administered via an intraperitoneal catheter	NCT03585764
Osteosarcoma	Seattle Children’s Hospital/Umoja BioPharma	Fluorescein-specific (universal) CAR T-cells plus folate–fluorescein conjugate	NCT05312411
Ovarian, NSCLC ^1^ and RCC ^2^	Instil Bio	Tumour-infiltrating lymphocytes engineered to express a FRα-specific CD28 + CD40 co-stimulatory receptor	NCT05397093

^1^ Non-small cell lung cancer; ^2^ renal cell cancer.

**Table 8 biology-12-00287-t008:** Ongoing CAR T-cell clinical trials directed against IL13Rα2 (assessed 3 November 2022).

Disease	Sponsor	Notes	Identifier
Glioma	Xuanwu Hospital	Patients may also receive anti-PDL1 antibody	NCT03423992
Paediatric malignant brain tumours	City of Hope Medical Center	Systemic lymphodepletion followed by intraventricular CAR T-cells	NCT04510051
Various adult brain tumours	City of Hope Medical Center	Intraventricular CAR T-cells	NCT04661384
Malignant glioma	CellabMED	Intravenous delivery	NCT05540873
Melanoma	Jonsson Comprehensive Cancer Center	Intravenous delivery post lymphodepletion	NCT04119024
Glioblastoma	City of Hope Medical Center	Intracranial CAR T-cells administered with systemic nivolumab (anti-PD1) and Ipilimumab (anti-CTLA-4)	NCT04003649

**Table 9 biology-12-00287-t009:** Ongoing CAR T-cell clinical trials directed against PSMA (assessed 2 November 2022).

Disease	Sponsor	Notes	Identifier
PSMA+ tumours	Shenzen Geno-Immune Medical Institute	Includes cases in which PSMA is demonstrated in tumour stroma	NCT04429451
Tumours that co-express PSMA and GD2	Shenzen Geno-Immune Medical Institute	Bispecific CAR T-cells	NCT05437315
Tumours that co-express PSMA and CD70	Shenzen Geno-Immune Medical Institute	Bispecific CAR T-cells	NCT05437341
mCRPC ^1^	Affil. Hospital of Xuzhou Medical University	Administered in combination with IL-2	NCT05354375
mCRPC and salivary gland cancer	Poseida	Transgenes delivered using PiggyBac. Vector also contains iCaspase-9 suicide gene	NCT04727151
mCRPC	Zhejiang University	Non-viral gene transfer. Vector includes PD1 inhibitory system.	NCT04768608
mCRPC	AvenCell Europe	Universal CAR in combination with PSMA-specific antibody derivative	NCT04633148
mCRPC	Tmunity	CAR containing CD2 co-stimulatory domain. Dual armouring with dnTGF-βR and PD1/CD28 switch receptor	NCT05489991
Sarcomas	Shenzen Geno-Immune Medical Institute	Multiple CAR T-cells combined targeting antigens that include PSMA	NCT04433221
Neuroblastoma	Shenzen Geno-Immune Medical Institute	CAR also targets GD2 and CD276	NCT04637503

^1^ Metastatic castrate resistant prostate cancer.

**Table 10 biology-12-00287-t010:** Ongoing CAR T-cell clinical trials directed against GD2 (assessed 25 November 2022).

Disease	Sponsor	Notes	Identifier
GD2+ Brain Tumours	Baylor College of Medicine	Armoured with constitutively active IL-7 receptor	NCT04099797
GD2+ non-brain Tumours	Baylor College of Medicine	Armoured with constitutively active IL-7 receptor	NCT03635632
Tumours that co-express GD2 and CD70	Shenzen Geno-Immune Medical Institute	Bispecific CAR T-cells	NCT05438368
Tumours that co-express GD2 and CD56	Shenzen Geno-Immune Medical Institute	Bispecific CAR T-cells	NCT05437328
Tumours that co-express GD2 and PSMA	Shenzen Geno-Immune Medical Institute	Bispecific CAR T-cells	NCT05437315
Neuroblastoma Osteosarcoma	UNC Lineberger Comprehensive Cancer Center	Armoured with IL-15 and inducible caspase-9	NCT03721068
Neuroblastoma	Shenzen Geno-Immune Medical Institute	Trispecific CAR T-cells directed against GD2, PSMA and CD276 (B7-H3)	NCT04637503
B-NHL ^1^	7th Affil. Hospital of Sun Yat-sen University	One of multiple targets	NCT04429438
Lung Cancer	UNC Lineberger Comprehensive Cancer Center	Armoured with IL-15 and inducible caspase-9	NCT05620342
DIPG/DMG ^2^	Crystal Mackall MD	Dasatinib-containing culture system [109]	NCT04196413
GD2+ tumours	Bambino Gesu Hospital and Research Institute	Armoured with inducible caspase-9	NCT03373097
Neuroblastoma Osteosarcoma	National Cancer Institute	Dasatinib-containing culture system	NCT04539366
Breast Cancer	7th Affil. Hospital of Sun Yat-sen University	One of multiple targets. GD2 considered a breast cancer stem cell marker	NCT04430595
Neuroblastoma	Baylor College of Medicine	IL-15 armoured NKT-cells	NCT03294954
Glioma	Xuanwu Hospital	One of multiple targets	NCT03423992
Sarcomas	Shenzen Geno-Immune Medical Institute	One of multiple targets	NCT03356782
Sarcomas	Shenzen Geno-Immune Medical Institute	One of multiple targets. Combination treatment with chemotherapy and/ or tumour vaccines	NCT04433221

^1^ B-cell non-Hodgkin’s Lymphoma; ^2^ diffuse intrinsic pontine glioma/diffuse midline glioma.

**Table 11 biology-12-00287-t011:** Ongoing CAR T-cell clinical trials directed against B7-H3 (assessed 25 November 2022).

Disease	Sponsor	Notes	Identifier
Glioblastoma	UNC Lineberger Comprehensive Cancer Center of Medicine	Intraventricular infusion	NCT05366179
Ovarian cancer	UNC Lineberger Comprehensive Cancer Center of Medicine	Intraperitoneal infusion	NCT04670068
Paediatric CNS tumours	Seattle Children’s Hospital	Intracranial delivery. Methotrexate selection system for CAR T-cells. Reference [113]	NCT04185038
Solid tumours	2nd Affil. Hospital of Guangzhou Medical University	One of multiple targets	NCT03198052
Glioblastoma	Beijing Tiantan Hospital		NCT05241392
Paediatric solid tumours	St. Jude’s Children’s Research Hospital		NCT04897321
Solid tumours in children and young adults	Seattle Children’s Hospital	In one arm, patients receive T-cells that co-express the B7-H3 CAR with a CD19 CAR in an effort to increase expansion and persistence. Reference [114]	NCT04483778
Hepatocellular carcinoma	Affil. Hospital of Xuzhou Medical University	Transhepatic arterial infusion	NCT05323201
Ovarian carcinoma	Affil. Hospital of Xuzhou Medical University	Intraperitoneal infusion	NCT05211557
Glioblastoma Multiforme	Crystal Mackall MD	Locoregional delivery	NCT05474378
Solid tumours	Shenzen Geno-Immune Medical Institute		NCT04432649
Glioblastoma	2nd Affil. Hospital School of Medicine, Zhejiang University	Intracerebral CAR T-cells are administered between temozolomide cycles	NCT04077866
Solid tumours	1st Peoples Hospital of Lianyungang		NCT05515185
Lung cancer TNBC ^1^	2nd Affil. Hospital of Guangzhou Medical University	Also targeted against EGFR	NCT05341492
Melanoma Lung cancer Colorectal cancer	4th Hospital of Hebei Medical University		NCT05190185
Pancreatic cancer	Shenzhen University General Hospital		NCT05143151
Neuroblastoma Osteosarcoma Gastric and Lung cancer	PersonGen BioTherapeutics	Intravenous and intra-tumoural delivery	NCT04864821
Neuroblastoma	Shenzen Geno-Immune Medical Institute	One of multiple targets	NCT04637503
	Baylor College of Medicine	IL-15 armoured NKT-cells	NCT03294954
Glioma	Xuanwu Hospital	One of multiple targets	NCT03423992
Sarcomas	Shenzen Geno-Immune Medical Institute	One of multiple targets	NCT03356782
Sarcomas	Shenzen Geno-Immune Medical Institute	One of multiple targets. Combination treatment with chemotherapy and/or tumour vaccines	NCT04433221

^1^ Triple-negative breast cancer.

**Table 12 biology-12-00287-t012:** Ongoing CAR T-cell clinical trials directed against Glypican-3 (assessed 13 December 2022).

Disease	Sponsor	Notes	Identifier
Childhood tumours that express Glypican-3	Baylor College of Medicine	IL-15 armoured CAR T-cells iCaspase-9 suicide gene	NCT04377932
Hepatocellular carcinoma	Baylor College of Medicine	IL-15 armoured CAR T-cells iCaspase-9 suicide gene	NCT05103631
Hepatocellular carcinoma	Affil. Nanjing Drum Tower Hospital		NCT04121273
Glypican-3+ tumours	SOTIO	Glutamic oxaloacetic transaminase 2 armouring [122]	NCT05354375
Hepatocellular carcinoma	National Cancer Institute		NCT05003895
Hepatocellular carcinoma	Tongji University		NCT05070156
Hepatocellular carcinoma	2nd Affiliated Hospital Guangzhou Medical University	In CD4+ T-cells, CAR also targets TGF-β and secretes IL-7/CCL19 and scFvs against PD1, CTLA4 and Tigit. In CD8+ T-cells, a Glypican-3/Dap10 CAR is expressed and both PD1 and HPK are knocked down	NCT03198546

**Table 13 biology-12-00287-t013:** Ongoing CAR-based clinical trials directed against NKG2D ligands (assessed 14 December 2022).

Disease	Sponsor	Notes	Identifier
NKG2D ligand+ tumours	Fudan University		NCT05131763
Liver metastatic colorectal carcinoma	3rd Affiliated Hospital Guangzhou Medical University	Hepatic artery infusion	NCT05248048
NKG2D ligand+ tumours	Jianming Xu		NCT05382377
NKG2D ligand+/ Claudin 18.2+ tumours	Jianming Xu	Also targets Claudin-18.2	NCT05583201

**Table 14 biology-12-00287-t014:** Ongoing CAR T-cell clinical trials directed against PSCA (assessed 14 December 2022).

Disease	Sponsor	Notes	Identifier
Prostate cancer	City of Hope Medical Center		NCT03873805
Prostate cancer	Bellicum Pharmaceuticals	Incorporates rimiducid inducible co-stimulatory domain	NCT02744287

**Table 15 biology-12-00287-t015:** Ongoing CAR T-cell clinical trials directed against CEA (assessed 14 December 2022).

Disease	Sponsor	Notes	Identifier
Colorectal cancer	Changhai Hospital		NCT05240950
CEA+ cancer	Chongqing Precision Biotech Co.		NCT05538195
CEA+ cancer	Chongqing Precision Biotech Co.		NCT04348643
CEA+ cancer	Chongqing Precision Biotech Co.		NCT05538195
CEA+ cancer	Weijia Fang	Intravenous or intraperitoneal administration	NCT05396300

**Table 16 biology-12-00287-t016:** Ongoing CAR T-cell clinical trials directed against CEA (assessed 15 December 2022).

Disease	Sponsor	Notes	Identifier
CD70+ cancer	Chongqing Precision Biotech Co.	Intravenous or intraperitoneal delivery	NCT05468190
CD70+ renal cell carcinoma	Chongqing Precision Biotech Co.		NCT05420519
CD70+ cancer	Chongqing Precision Biotech Co.	Intravenous or intraperitoneal delivery	NCT05420545
CD70+ cancer	Weijia Fang	Intravenous or intraperitoneal delivery	NCT05518253
CD70+ clear cell renal carcinoma	CRISPR Therapeutics	Allogeneic CRISPR-Cas9 engineered T-cells	NCT04438083
CD70+ cancer	National Cancer Institute	Targeted using CD27	NCT02830724

**Table 17 biology-12-00287-t017:** Ongoing CAR T-cell clinical trials directed against CD133 (assessed 15 December 2022).

Disease	Sponsor	Notes	Identifier
Glioma	Xuanwu Hospital, Beijing	One of multiple targets; administered with or without anti-PD1	NCT03423992
Sarcomas	Shenzhen Geno-Immune Medical Institute	One of multiple targets	NCT03356782

**Table 18 biology-12-00287-t018:** Ongoing CAR T-cell clinical trials directed against EpCAM (assessed 7 January 2022).

Disease	Sponsor	Notes	Identifier
Solid tumours	Sichuan University	Nasopharyngeal, breast and gastric cancers main focus	NCT02915445
Gastrointestinal cancers	Zhejiang University	Hepatocellular, colorectal, gastric and pancreatic cancers	NCT05028933

## Data Availability

Not applicable.

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
