# Peer review of "CAR Based Immunotherapy of Solid Tumours—A Clinically Based Review of Target Antigens"

_biology, 2023, doi:10.3390/biology12020287_

Round 1

Reviewer 1 Report

The authors summarized clinical data for studies involving 30 of targets in solid tumor CAR-based immunotherapy and listed ongoing clinal trails, highlighting challenges.  The authors also discussed several new directions of CAR-based immunotherapy. 

comment:

Can authors give more details clinical or research information about PD-L1 based CAR T immunotherapy?

Author Response

We thank reviewer 1 for this comment. The section on PD-L1-targeted CAR T-cells has been expanded and now reads as follows:

Programmed death receptor ligand 1 is widely expressed on a range of solid tumours and has been successfully targeted using CAR T-cells in a number of pre-clinical studies [119,120]. A further theoretical attraction of this approach is the demonstration that PD-L1-targeted CAR T-cells can promote the upregulation of this target on PD-L1 low tumour cells [121]. Balancing this, it was also noted that amplified PD-L1 expression in normal tissues could have detrimental effects due to impairment of PD-L1-dependent tolerance mechanisms in vital organs. Natural killer cells have also been engineered to express a PD-L1 specific CAR and effectively killed a broad range of tumour cell lines [122] while also eliminating immunosuppressive myeloid cells that express high levels of PD-L1 [123]. However, a CAR T-cell clinical trial in NSCLC was terminated in China owing to an undisclosed serious adverse event (NCT03330834, accessed 7.1.2023) which responded to treatment with tocilizumab and steroids [121]. One other clinical study has safely employed a PD1/ CD28 switch receptor, which was co-expressed with a CD19-specific 4-1BB-containing CAR [124].

Reviewer 2 Report

This manuscript reviews the progress of 20 kinds CAR T-cell clinical trials. It is the newest and perfect. 

There are 4 references had better been changed such as 34, 38 ,78 and 124.

Author Response

We thank reviewer 2 for their comments.

Reference 34, 37 and 78 have been removed.

We have also reviewed reference 124 (Hickman et al) and would like to maintain this reference since it provides expression data for GPC3 in normal tissue which we believe is relevant to its potential tractability as a CAR target.

Reviewer 3 Report

This is a comprehensive review describing clinical outcome of Chimeric antigen receptor (CAR)-based immunotherapy on solid cancer. The manuscript is well-written and well-organized and deserves publication. Below are some minor comments that need to be addressed before the manuscript can be published.

Line 38: It would be good if the authors could present a Figure illustrating the genetic engineered CAR.

Line 69: How does human scFv reduce immunogenicity of the CAR? Please describe.

Line 83: Please mention the antigen towards the pembrolizumab is reacting. The same for traztuzumab in line 132, pertuzumab in line 187, and tocilizumab in line 244, nivolumab and ipilimumab in Table 8.

Lines 173 and 179: There are empty spaces. Maybe intended to be an indent?

Line 511: Please explain the concept "mutein".

The right margin should be justified in some of the sections.

Author Response

We thanks reviewer 3 for their comments.

  1. A figure has been inserted as suggested.
  2. Text has been modified to read: One patient died of sepsis and anti-CAR antibodies were detected in a number of subjects, presumably due to the inclusion of a murine scFv in the CAR ectodomain. To address this, a human scFv was next incorporated to reduce immunogenicity of the CAR.
  3. Targets of monoclonal antibodies have been added.
  4. Mutein has been explained as mutated cytokine.
  5. Spacing and justification of text has been re-checked.